Honey bee viruses in Serbian colonies of different strength

Cirkovic Dragan 1
Stevanovic Jevrosima rocky@vet.bg.ac.rs 2
Glavinic Uros 2
Aleksic Nevenka 3
Djuric Spomenka 4
Aleksic Jelena 5
Stanimirovic Zoran 2
1 Department of Chemical and Technological Sciences, State University of Novi Pazar , Novi Pazar , Serbia
2 Department of Biology, Faculty of Veterinary Medicine, University of Belgrade , Belgrade , Serbia
3 Department of Parasitology, Faculty of Veterinary Medicine, University of Belgrade , Belgrade , Serbia
4 Department of Economics and Statistics, Faculty of Veterinary Medicine, University of Belgrade , Belgrade , Serbia
5 Institute of Molecular Genetics and Genetic Engineering (IMGGE), University of Belgrade , Beograd , Serbia
Gillespie Joseph
Electronic publication date: 2018 Nov 14
Publication date: 2018
Volume: 6
Electronic Location ID: e5887
Received 2018 Jun 21; Accepted 2018 Oct 7
Copyright: ©2018 Cirkovic et al.
Copyright year: 2018
Copyright holder: Cirkovic et al.
License: This is an open access article distributed under the terms of the Creative Commons Attribution License, which permits unrestricted use, distribution, reproduction and adaptation in any medium and for any purpose provided that it is properly attributed. For attribution, the original author(s), title, publication source (PeerJ) and either DOI or URL of the article must be cited.
License URL: https://creativecommons.org/licenses/by/4.0/

Keywords: Apis mellifera, Viruses, Colony strength, Phylogeny, Beekeeping, Bee pathology

Funding: Ministry of Education, Science and Technological Development of the Republic of Serbia III 46002 This work was supported by the Ministry of Education, Science and Technological Development of the Republic of Serbia (grant number III 46002, led by Prof. dr Zoran Stanimirović). The funders had no role in study design, data collection and analysis, decision to publish, or preparation of the manuscript.

==============================
Protection of honey bees is of great economic importance because of their role in pollination. Crucial steps towards this goal are epidemiological surveys of pathogens connected with honey bee losses. In this study deformed wing virus (DWV), chronic bee paralysis virus (CBPV), acute bee paralysis virus (ABPV) and sacbrood virus (SBV) were investigated in colonies of different strength located in five regions of Serbia. The relationship between colony strength and virus occurrence/infection intensity were assessed as well as the genetic relationship between virus sequences from Serbia and worldwide. Real-time RT-PCR analyses detected at least one virus in 87.33% of colonies. Single infection was found in 28.67% colonies (21.33%, 4.00%, 2.67% and 0.67% in cases of DWV, ABPV, SBV and CBPV, respectively). In the majority of colonies (58.66%) more than one virus was found. The most prevalent was DWV (74%), followed by ABPV, SBV and CBPV (49.30%, 24.00% and 6.70%, respectively). Except for DWV, the prevalence of the remaining three viruses significantly varied between the regions. No significant differences were found between colony strength and either (i) the prevalence of DWV, ABPV, SBV, CBPV and their combinations, or (ii) DWV infection levels. The sequences of honey bee viruses obtained from bees in Serbia were 93–99% identical with those deposited in GenBank.

Introduction

Honey bees (Apis mellifera) are well-known beneficial insects for their popular products, and much more for their important role in pollination (Venturini et al., 2017). Unfortunately, huge losses of managed honey bee colonies were reported worldwide (Van Engelsdorp et al., 2008; Van Engelsdorp et al., 2009; Bacandritsos et al., 2010; Van Engelsdorp et al., 2012; Lee et al., 2015; Antúnez et al., 2016; Kulhanek et al., 2017; Brodschneider et al., 2018), but no single factor was confirmed to be a certain cause of colony mortality (Van Engelsdorp et al., 2009), although the mite Varroa destructor and associated viruses have most often been cited (Francis, Nielsen & Kryger, 2013; McMenamin & Genersch, 2015; Steinhauer et al., 2018).

More than 22 honey bee viruses have been identified and described so far (Genersch, 2010) which exist or co-exist in individual bees or colonies, but may remain unnoticed (Chen & Siede, 2007; Brutscher, McMenamin & Flenniken, 2016). However, several viruses transferred by V. destructor considered to pose increasing risk to colonies’ health (Martin et al., 2012) including deformed wing virus (DWV), chronic bee paralysis virus (CBPV), acute bee paralysis virus (ABPV) and the sacbrood virus (SBV), all of them seeming to have worldwide occurrence and distribution (Genersch, 2010; Simeunović et al., 2014a; Brutscher, McMenamin & Flenniken, 2016).

In Southeastern Europe, the presence and prevalence of bee viruses have been investigated in Hungary, Slovenia and Croatia (Bakonyi et al., 2002; Forgách et al., 2008; Toplak et al., 2012; Tlak Gajger et al., 2014; Tlak Gajger, Bičak & Belužić, 2014). In Serbia, four honey bee viruses were reported: ABPV, Egypt bee virus J strain (EBV), cloudy wing virus (CWV) and the black queen cell virus (BQCV) by Kulinčević, Ball & Mladjan (1990), and DWV and ABPV by Simeunović et al. (2014a). However, due to the lack of information on the prevalence of SBV and CBPV, as well as the long time which passed since the previous investigations necessity demands newer research.

There is limited information about the relation between colony strength and presence of bee viruses.

The present study was aimed at: (1) surveying the prevalence of DWV, SBV, ABPV and CBPV in honey bee colonies of different strength in Serbia; (2) exploring the differences between virus prevalence/intensity of infection and colony strength; and (3) phylogenetic analyses to reveal the relationship between viruses found in Serbia and those deposited in GenBank.

Material and Methods

One hundred and fifty colonies were sampled from 32 apiaries (approximately five colonies per apiary) located in five administrative regions of Serbia (Fig. 1) in autumn (in period from September 25 to October 5) 2017.

Figure 1 Prevalence patterns of investigated viruses in honey bee colonies in five regions of Serbia.

On each apiary two strong, one medium and two weak colonies were chosen. Colony strength assessment and classification were done as in Cavigli et al. (2016). The selected colonies were without visible signs of any disease.

About a hundred workers, both foragers and house bees were chosen for each sample, placed in sterile test tubes on dry ice, and stored at −20 °C until being processed.

Thirty randomly selected specimens taken from each bee sample were pulverized and homogenized in 5 mL of PBS solution. After centrifugation, from 140 µL of the supernatant the RNA was extracted with ZR Viral RNA Kit™ (Zymo Research, Orange, CA, USA).

The obtained sequences were amplified in Rotor-Gene Q 5plex (Qiagen, Hilden, Germany) and the target viruses detected with the Rotor-Gene Probe RT-PCR Kit (Qiagen, Germany), in separate single-step reactions. The primer pairs and probes for DWV, ABPV and SBV (Table 1) were the same as used by Chantawannakul et al. (2006) and for CBPV those deployed by Blanchard et al. (2007). The final primer concentration 800 nM and probe concentration of 400 nM proved optimum. The analyses were done in conditions defined in the work of Simeunović et al. (2014a). With each set of sample reactions, standard dilutions of the control sample were run and a threshold level set according to the standard curve obtained.

Table 1 Primers and probes (TaqMan Probe®) used for RNA molecular identification of investigated viruses in real-time RT-PCR.

Primer/Probe
name	Sequence	Primer	Virus	Primer/ Probe authors	
DWV958F
DWV9711R
DWV9627T	5′-AAATTCTCTCACAGTCCAAG-3′
5′CAACAGGTAATTTTCCTTTAG-3′
5′-CATGCTCGAGGATTGGGTCGTCGT-3′	Forward
Reverese
Probe	Deformed Wing Virus	Chantawannakul et al. (2006)	
APV95F
APV159R
APV121T	5′- TCCTATATCGACGACGAAAGACAA-3′
5′- GCGCTTTAATTCCATCCAATTGA-3′
5′- TTTCCCCGGACTTGAC-3′	Forward
Reverese
Probe	Acute Bee Paralysis Virus	Chantawannakul et al. (2006)	
SBV311F
SBV380R
SBV331T	5′-AAGTTGGAGGCGCGYATTTG-3′
5′-CAAATGTCTTCTTACDAGAAGYAAGGATTG- 3′
5′-CGGAGTGGAAAGAT-3′	Forward
Reverese
Probe	Sacbrood Virus	Chantawannakul et al. (2006)	
CBPV1F
CBPV2R
CBPVT	5′-CGCAAGTACGCCTTGATAAAGAAC-3′
5′-ACTACTAGAAACTCGTCGCTTCG-3′
5′-TCAAGAACGAGACCACCGCCAAGTTC-3′	Forward
Reverese
Probe	Chronic Bee Paralyses Virus	Blanchard et al., (2007)	

Selected RNA isolates were subjected to endpoint RT-PCRs using primer pairs and following the recommendations from ANSES (2011). The sequencing of each amplicon was done in both orientations in ABI 3130 Genetic Analyzer (Applied Biosystems, Foster City, CA, USA).

The obtained partial nucleotide (nt) sequences of honeybee viruses were identified by the BLAST search (http://blast.ncbi.nlm.nih.gov/Blast.cgi) against the GenBank database. Sequences encoding a partial coding sequence (cds) of polyprotein gene of DWV, a capsid protein gene of ABPV, a partial cds of RNA-dependent RNA polymerase (RdRp) gene of CBPV, and a partial cds of polyprotein gene of SBV were recovered. They were used for phylogenetic analyses along with related sequences deposited in GenBank. The best models of sequence evolution according to the Bayesian Information Criterion assessed MEGA version 6 (Tamura et al., 2007) were as follows: T92 + G for DWV and SBV, T92 + I for ABPV, and K2 + G + I for CBPV. Evolutionary relations assessed using these models of sequence evolution and the Neighbor-Joining (NJ) algorithm were shown as phylograms. Statistical support was tested with 1,000 nonparametric bootstrap (BS) replicates, with 50% ≥ BS ≤ 74% considered moderate support, and BS ≥ 75% considered good support.

Statistical analysis

Depending on data characteristics (testing for normality), the results were presented through the mean and standard deviation, or the median and interquartile range were used. Differences were tested using ANOVA, t-test, or, where appropriate, non-parametric Mann–Whitney U test and Kruskal–Wallis test. Pearson chi-square analysis (or Fisher’s exact test) were applied where necessary. Data analysis was performed using IBM SPSS Statistics ver. 21.0 software (IBM, Armonk, NY, USA).

Results

All of the four viruses in examined samples of adult bees were detected. Their prevalence differed depending on the region of sampling (Fig. 1). In 150 honey bee samples (colonies), the prevalence of these four viruses was as follows: 74% of DWV, 49.30% of ABPV, 24.00% of SBV and 6.70% of CBPV. Samples negative for all four viruses comprised 12.67% of the colonies investigated (Fig. 2).

Figure 2 Overall prevalence of CBPV, SBV, ABPV and DWV in Serbian bees (analyzed in 150 samples).

In 87.33% of samples analysed at least one virus was detected. Single infection was found in 28.67% of colonies (DWV, ABPV, SBV and CBPV in 21.33%, 4.00%, 2.67% and 0.67% colonies, respectively, Table 2). The majority of colonies (58.66%) were found to be infected with more than one virus. DWV had the highest prevalence in all regions (66.70–83.30%), while the least prevalent virus was CBPV (0–19%). Except for DWV, the prevalence of the remaining three viruses was significantly different between different regions (χ2 test: DWV P = 0.554, ABPV P = 0.001, SBV P < 0.001, CBPV P = 0.030).

Table 2 Prevalences of single and simultaneous virus infections in honey bee samples from Serbia.

No. of viruses in simultaneous infection	Type of infection	No. of samples	%	
0	/	19	12.67	
1	DWV	32	21.33	
ABPV	6	4.00	
SBV	4	2.67	
CBPV	1	0.67	
2	DWV, ABPV	44	29.33	
DWV, CBPV	6	4.00	
DWV, SBV	11	7.33	
ABPV, SBV	11	7.33	
3	DWV, SBV, ABPV	12	8.00	
DWV, ABPV, CBPV	3	2.00	
4	DWV, ABPV, SBV, CBPV	1	0.67	
Notes.

DWV deformed wing virus

CBPV chronic bee paralysis virus

ABPV acute bee paralysis virus

SBV sacbrood virus

The prevalence of each virus in weak, medium and strong colonies is shown in Fig. 3. DWV was most prevalent in strong colonies (78%), followed by weak (72.90%) and medium colonies (68.80%). The highest number of ABPV-positive samples was recorded in medium colonies (62.50%), followed by strong (49.20%) and weak colonies (42.40%). SBV was found in 25.40% of weak colonies, 25.00% of medium ones and in 22.00% of strong colonies. CBPV were found in 9.40% of medium colonies, 8.50% weak and 3.40% strong colonies. No significant differences were recorded in the prevalence of DWV, ABPV, SBV and CBPV infections (and their combinations) between weak, medium and strong colonies were recorded with the χ2 test (Fig. 3). The significance of differences in virus infection levels (expressed through Ct values) between colonies of different strength were also tested. In order to avoid any confounding factor originated from the presence of other viruses in multiple infections, only single infections were taken into consideration in the data analysis. The numbers of samples with single ABPV, SBV and CBPV infections were not statistically valid for the comparison of their Ct values in strong, medium and weak colonies; therefore, only DWV infection intensity was eligible for testing in respect to colony strength. No significant differences were found between DWV infection levels in colonies of different strength (Fig. 4; ANOVA, F = 0.681, P = 0.513). In addition, the results presented in Fig. 5 show that DWV Ct values significantly differ between single DWV infection and double infections caused by DWV and ABPV, SBV or CBPV (ANOVA, F = 7.510, P < 0.001).

Figure 3 Prevalence of DWV, ABPV, SBV, CBPV and their combinations (A–K) in weak, medium and strong colonies.

Figure 4 Intensity of DWV infection (Ct values) in weak, medium and strong colonies.

Figure 5 Intensity of virus infections (Ct values) in single DWV infection and in double infections (DWV+ABPV, DWV+CBPV and DVW+SBV).

Phylogenetic analyses

Phylogenetic trees showing evolutionary relations between Serbian and worldwide honeybee viruses DWV, ABPV, SBV and CBPV are shown in Figs. 6–9, respectively.

Figure 6 Neighbour-Joining tree of studied DWV sequences.

The tree was constructed using a 420 nt long aligned matrix of 18 sequences encoding a partial coding sequence (cds) of polyprotein gene of DWVs. VDVs (AY251269 and JF440525) were used as outgroups to root the tree. Viruses are indicated with GenBank Access. Nos. and the country of origin. Numbers at nodes represent bootstrap support. Bar on the left shows the number of nucleotide substitutions per site.

Nine DWV sequences detected in Serbian honeybees were deposited in GenBank: Serbia D1 (KM001902); Serbia D2 (KM001903); Serbia D3 (KM001904); Serbia D4 and D5 (KM001905, these two sequences were identical and thus they were deposited in the GenBank under the same accession number); Serbia D6 (KM001906); Serbia D7 (KM001907); Serbia D8 (KM001908); and Serbia D9 (KM001909). BLAST search found 99 to 98% nucleotide identities with DWV sequences in the database. Eighteen additional DWVs sequences from the GenBank were used for phylogenetic analysis, and VDVs (AY251269 and JF440525) were used as outgroups to root the tree. The length of the aligned matrix was 420 nt. Evolutionary relations of studied DWVs are shown in Fig. 6. Five Serbian DWVs organized into two moderately supported clusters, comprising three and two sequences, respectively, were closely related to DWVs from the United Kingdom, while others were dispersed throughout the tree.

Two Serbian ABPV sequences, KL4 and KL5, were identical, and thus they were deposited in the GenBank under the same accession number (KM001899). They showed 97 to 93% nucleotide identities to ABPV sequences in the database. Ten additional European ABPV sequences from the GenBank were used for phylogenetic analysis, and KBV (AY452696) was used as outgroup to root the tree shown in Fig. 7. The length of the aligned matrix was 398 nt. Serbian ABPVs were closely related to the Hungarian ones while Western and Northern European viruses formed separate clusters.

Figure 7 Neighbour-Joining tree of studied ABPV sequences.

The tree was constructed using a 398 nt long aligned matrix of 12 sequences encoding a capsid protein of ABPVs. KBV (AY452696) was used as outgroups to root the tree. Viruses are indicated with GenBank Access. Nos. and the country of origin. Numbers at nodes represent bootstrap support. Bar on the left shows the number of nucleotide substitutions per site.

Three identical Serbian SBV sequences, S1, KL2 and KL25, deposited in the GenBank under the same accession number, KM001901, showed 99 to 94% sequence identity rates with other SBVs in the database. Seventeen additional SBVs from the GenBank were used for phylogenetic analysis. The length of the aligned matrix was 570 nt, and the recovered tree is shown in Fig. 8. Three Serbian SBVs cluster together with SBVs from the continental Europe.

Figure 8 Neighbour-Joining tree of studied SBV sequences.

The tree was constructed using a 429 nt long aligned matrix of 20 sequences encoding a partial coding sequence (cds) of polyprotein gene of SBVs. Viruses are indicated with GenBank Access. Nos. and the country of origin. Numbers at nodes represent bootstrap support. Bar on the left shows the number of nucleotide substitutions per site

Figure 9 Neighbour-Joining tree of studied CBPV sequences.

The tree was constructed using a 570 nt long aligned matrix of 12 sequences encoding a partial coding sequence (cds) of RNA-dependent RNA polymerase (RdRp) gene of CBPV. FJ345326 was used as outgroup to root the tree. Viruses are indicated with GenBank accession numbers and the country of origin. Numbers at nodes represent bootstrap support. Bar on the left shows the number of nucleotide substitutions per site.

Two identical Serbian CBPV sequences, CBPV-1 and CBPV-3, deposited in the GenBank under the same accession number (KM001900) show 96 to 93% sequence identity with other CBPVs in the GenBank. The length of the aligned matrix comprising Serbian and ten additional CBPVs was 429 nt. The relations of studied CBPVs are shown in Fig. 9.

Discussion

In the era of intensive agriculture and serious decline in pollinator populations worldwide, primarily honey bees (Goulson et al., 2015), it is of great importance to gain an insight into the distribution and prevalence of factors most often connected with bee losses in any geographic region (Van Engelsdorp et al., 2008; Van Engelsdorp et al., 2009; Van Engelsdorp et al., 2012; Cavigli et al., 2016). In this study, samples from clinically healthy colonies in Serbian apiaries were analysed by real-time RT-PCR in order to detect honey bee viruses (DWV, ABPV, SBV and CBPV) and determine their prevalence patterns and prevalence. The results revealed DWV to be the most prevalent virus in Serbian apiaries, not unlike in many other countries: Hungary (Bakonyi et al., 2002), France (Tentcheva et al., 2004; Mouret et al., 2013), Austria (Berenyi et al., 2006), Slovenia (Toplak et al., 2012) and Uruguay (Giacobino et al., 2016). High prevalence of DWV (74%) and ABPV (49.3%) recorded in Serbian apiaries are not surprising, knowing their close relation to V. destructor mite infestation and their persistence as subclinical infection in apparently healthy colonies (Gauthier et al., 2007; Mouret et al., 2013; Wells et al., 2016). The average prevalence of SBV in Serbian samples was 24%, and none of the investigated colonies exhibited signs of sacbrood disease. The absence of disease signs in all recorded SBV-positive colonies may be the result of prominent hygienic behaviour (Swanson et al., 2009), previously confirmed for honey bees throughout Serbia (Stanimirović et al., 2002; Stanimirović, Stevanović & Ćirković, 2005; Stanimirović et al., 2008; Stanimirović et al., 2011). The frequency of SBV in Serbia is similar with 40.24% recorded in Croatia (Tlak Gajger et al., 2014), considerably lower than 86% from France (Tentcheva et al., 2004) and 100% from Uruguay (Antúnez et al., 2006), but several times higher than 1.1%, 1.4% and 2% reported in Spain (Antúnez et al., 2012), England (Baker & Schroeder, 2008) and Hungary (Forgách et al., 2008), respectively. Low prevalence of CBPV in the samples is typical for asymptomatic colonies (Tentcheva et al., 2004). The rate of 0–19% CBPV-positive samples affirmed in Serbia is in accordance with the results obtained in the majority of Austrian federal states (Berenyi et al., 2006), Chinese provinces (Ai, Yan & Han, 2012), Korea (Choe et al., 2012), Slovenia (Toplak et al., 2012), and the apiaries from Denmark (Nielsen, Nicolaisen & Kryger, 2008) and France (Tentcheva et al., 2004).

Among monitored honey bee viruses in Serbia, the highest incidence was recorded for DWV (66.7–83.3%). No significant differences in its prevalence among Serbian regions is not surprising knowing its global occurrence (Wilfert et al., 2016) and its dominance over other viruses in variable environmental conditions (Giacobino et al., 2016).

The second most common virus in Serbian apiaries was ABPV, but its incidence (16.7–68.2%) significantly varied between the regions. The prevalence of SBV and CBPV also displayed dissimilar patterns in environmentally different regions. Additional investigations are necessary to explain the observed significant differences. It can be assumed that these results may reflect the beekeepers’ negligence of apicultural measures (Stanimirović et al., 2007a), but also may have risen from different means of V. destructor control (Nielsen, Nicolaisen & Kryger, 2008), which may be the reason only in ABPV infection, since not all viruses are transmitted by varroa mites (Glenny et al., 2017).

Nevertheless, differences in orographic factors and forage quality between regions should be also considered as the environment was suggested as a key factor interacting with local bee populations and ecogenotypes (Stanimirović, Stevanović & Ćirković, 2005; Giacobino et al., 2016). Our results concerning 87.33% samples with at least one virus and 58.66% with two or more are similar to those observed in Austria (Berenyi et al., 2006), France (Tentcheva et al., 2004; Gauthier et al., 2007) and Slovenia (Toplak et al., 2012).

Interestingly, no significant differences were found in the presence of DWV, ABPV, SBV and CBPV infections (and their combinations) in colonies of various strength. In addition, no significant differences were affirmed between single DWV infection levels (expressed through a Ct value) in colonies of different strength. These results may speak in favour of crucial influence of predisposing factors—pathogens, parasites, poor-quality nutrition, pesticides, and unfavourable climate conditions—on bee vitality (Stanimirović et al., 2007a; Simeunovic et al., 2014b; Abbo et al., 2017; Annoscia et al., 2018; Glavinic et al., 2017; Stevanovic et al., 2016). Special emphasis should be put on the negative influence of infestation with V. destructor, a biological and mechanical vector of at least two viruses, DWV and APBV, (Ryabov et al., 2014; Abbo et al., 2017) and a possible factor that could contribute Nosema ceranae spreading (Glavinić et al., 2014). In addition, we may assume that bees highly infected with viruses do not return from the field committing “altruistic suicide" to regulate colony virus load as in cases of V. destructor and/or N. ceranae infected bees (Kralj & Fuchs, 2006; Higes et al., 2008). In our study, the presence of another virus(es), ABPV, SBV or CBPV, in co-infections significantly influenced the intensity of DWV infection. The observed differences in DWV Ct values between co-infections and single DWV infections could be explained with the influence of simultaneous replication of the another present virus, wherein the influence may be stimulatory or suppressive. However, we should have in mind recent characterization of DWV master variants (DWV-A, DWV-B, and DWV-C) and their impact on bee health (McMenamin & Flenniken, 2018).

Very small percentages of multiple infections in comparison with single infections found in this study point out the possibility that the former are related with severe V. destructor infestations commonly observed in Serbian apiaries (Stanimirović et al., 2007a; Stanimirović et al., 2017).

High identity rates among relatively short studied nucleotide sequences of DWVs account for the poorly supported and unresolved phylogenetic tree (Fig. 6). However, the observed close genetic distance between all DWVs is concordant with the hypothesis of their relatively recent evolutionary diversification and worldwide spread, potentially connected to the geographic expansion of their main vector, V. destructor (Berenyi et al., 2007; Wilfert et al., 2016). On the other hand, Serbian and Hungarian ABPVs are closely related, and this may be explained by the geographical vicinity and trade between beekeepers of the two countries. Both Serbian and Hungarian ABPVs are relatively distant from those from the Western and Northern Europe, and this finding is in accordance with the report of Bakonyi et al. (2002) that Hungarian ABPVs are not closely related with Western and Northern European ABPVs. Although Serbian SBVs are closely related with SBVs from the continental Europe, further analysis, involving sequences from neighbouring countries, are required for determining whether similar separation exists with SBVs, as it has been affirmed in case of Serbian ABPV. CBPVs from Serbia, France, Belgium and Spain are monophyletic but Serbian CBPVs occupy a rather long branch indicating a non-negligible genetic distance between Serbian and mentioned CBPVs. These findings may indicate that CBPV (which is taxonomically and genetically very different from the other three honey bee viruses analysed in this study), may have different epizootiological character, and hence, is less intensively involved in the geographical spread of honey bee virus strains (Ribière, Olivier & Blanchard, 2010). Alternatively, unique genetic properties (higher mutation rate or segment rearrangements) may explain the genetic seclusion of the Serbian CBPVs. However, for better understanding of viral diversity in honey bee colonies, additional analyses are needed. This is in accordance with the opinion of Galbraith et al. (2018), who also emphasized the importance of virus development dynamics and its possible impact on honey bees. Studies on bee pathogens causing colony decline in Serbia were mainly focused on Nosema sp. (Stanimirovic et al., 2007b; Stevanovic et al., 2011; Stevanovic et al., 2013; Glavinić et al., 2014; Simeunovic et al., 2014b) and V. destructor (Stanimirović et al., 2002; Stanimirović, Stevanović & Ćirković, 2005; Stanimirovic et al., 2005; Stevanovic et al., 2008; Stanimirović et al., 2011; Radakovic et al., 2013; Gajic et al., 2013; Glavinić et al., 2014; Stanimirović et al., 2017) with only one study dealing with bee viruses (Simeunović et al., 2014a). Therefore, our work represents an important contribution towards better understanding of bee pathogens in Serbia.

Conclusions

This work represents the first thorough investigation aimed at the constitution of the epidemiological baseline regarding molecular identification, prevalence patterns and prevalence of honey bee viruses in Serbia. The geographic origin and strength of honey bee colonies in Serbia proved to be insufficient to induce significant differences in the prevalence of the investigated viruses. Infection intensity of DWV presented through Ct value greatly depends on the presence of co-infection with other viruses. However, single ABPV, SBV and CBPV infections were not frequent enough to allow the comparison of their Ct values. In addition, the sequence analyses of Serbian honey bee viruses confirmed their identity and enabled an insight into their phylogenetic relationship with those found worldwide.

Supplemental Information

Supplemental Information 1 Virus occurence in different regions of Serbia

Region mark 1 North 2 East 3 South 4 West 5 Center Colony strength mark 3 Strong 2 Medium 1 Weak

Click here for additional data file.

Figure S1 Real-time RT-PCR amplification curves obtained with probes (TaqMan Probe®) showing the presence of single infections with DWV, ABPV, CBPV and SBV

Click here for additional data file.

Figure S2 Intensity of virus infections (Ct values) in single DWV infection and in multiple infections combinations (DWV+ABPV+CBPV; DWV+ABPV+SBV; DVW+ABPV+CBPV+SBV)

Click here for additional data file.

Table S1 Ct value ratios of viruses in double virus infections in colonies of different strength

Click here for additional data file.

Supplemental Information 2 Database of bee virus genome sequnces

Click here for additional data file.

Additional Information and Declarations

Competing Interests

Author Contributions

DNA Deposition

Data Availability

The authors declare there are no competing interests.

Dragan Cirkovic conceived and designed the experiments, analyzed the data, authored or reviewed drafts of the paper, approved the final draft.

Jevrosima Stevanovic analyzed the data, authored or reviewed drafts of the paper, approved the final draft.

Uros Glavinic performed the experiments, prepared figures and/or tables.

Nevenka Aleksic authored or reviewed drafts of the paper, approved the final draft.

Spomenka Djuric analyzed the data, prepared figures and/or tables.

Jelena Aleksic analyzed the data, prepared figures and/or tables, for second version of the paper, she made new phylogenetic trees, made new Figures 6, 7, 8 and 9 in better resolution and corrected text and Figure legends in connection with Phylogenetic analyses.

Zoran Stanimirovic conceived and designed the experiments, analyzed the data, contributed reagents/materials/analysis tools, authored or reviewed drafts of the paper, approved the final draft.

The following information was supplied regarding the deposition of DNA sequences:

The Deformed wing virus isolate Serbia D1 polyprotein gene, partial cds sequence described here are accessible via GenBank accession numbers KM001902.1.

The Deformed wing virus isolate Serbia D2 polyprotein gene, partial cds sequence described here are accessible via GenBank accession number KM001903.1.

The Deformed wing virus isolate Serbia D3 polyprotein gene, partial cds sequence described here are accessible via GenBank accession number KM001904.1.

The Deformed wing virus isolate Serbia D4 polyprotein gene, partial cds sequence described here are accessible via GenBank accession number KM001905.1.

The Deformed wing virus isolate Serbia D6 polyprotein gene, partial cds sequence described here are accessible via GenBank accession number KM001906.1.

The Deformed wing virus isolate Serbia D7 polyprotein gene, partial cds sequence described here are accessible via GenBank accession number KM001907.1.

The Deformed wing virus isolate Serbia D8 polyprotein gene, partial cds sequence described here are accessible via GenBank accession number KM001908.1.

The Deformed wing virus isolate Serbia D9 polyprotein gene, partial cds sequence described here are accessible via GenBank accession number KM001909.1.

The following information was supplied regarding data availability:

NCBI: KM001899.1–KM001909.1

University of Belgrade, Biology Department: https://biologija.vet.bg.ac.rs/uzgoj-i-nega-pcela/.

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
