# Peer review of "Honey bee viruses in Serbian colonies of different strength"

_PeerJ, doi:10.7717/peerj.5887_

## Round 0.1 · original submission · Minor Revisions

Dear Dr. Cirkovic and colleagues:

Thanks for submitting your manuscript to PeerJ. I have now received three independent reviews of your work, and as you will see, all are very favorable. Well done! Nonetheless, the reviewers raised some relatively minor concerns about the research, and areas where the manuscript can be improved. Please consider missing references, criticisms regarding data presentation and explanation (in figures), and possibly better-suited statistical tests. Please remember to use the marked-up manuscript kindly provided by reviewer 3.

I agree with the criticisms raised by the reviewers, and thus feel that their concerns should be adequately addressed before moving forward. Therefore, I am recommending that you revise your manuscript accordingly. I do believe that your manuscript will be ready for publication once these issues are addressed.

Good luck with your revision,

-joe

·

Basic reporting

no comment

Experimental design

no comment

Validity of the findings

no comment

Additional comments

Editors,

The manuscript entitled, “Honey bee viruses in Serbian colonies of different
strength (#28925)” by Cirkovic et al. describes virus incidence in Serbian honey bee colonies. This work provides information that is relevant to global honey bee virus epidemiology, as well as provides a baseline for future honey bee monitoring studies in this region. In general, this manuscript is well-written, and the figures are informative.

Points to clarify or address before publication include:

1. Figure 2 - the number of samples analyzed for each virus should be included in the figure (i.e., n= 300) – maybe below each bar or in the figure caption if all samples were tested for all viruses.

2. Figure 3 and Line 81 Defining what the authors mean by “occurrence” and “prevalence” will clarify the results; it seems abundance should be used for qPCR data.
See - Cavigli et al Apidologie 2015

3. Figure 4 – It is not clear what the main point of this qPCR figure is – maybe it should be a supplemental figure. Detection of a single virus in a sample doesn’t seem like a really important result.

4. Figure 5 – Ct values seem really high. The authors’ could include a supplemental figure or just describe the linear equation that described their qPCR assay and its linear range.
Why are just Ct values shown in this figure when Line 105 indicated that the ddCt method was used?

5. Figure 5 B This panel would be more informative in the qPCR data was included for both viruses. It is common for multiple viruses to be detected in any honey bee samples, so likely levels of infection (virus abundance) is more important than presence.

6. Figures 6, 7, 8, 9 - Accession numbers are required for publication; it looks like there are in the text, but not in the figure captions. The caption should include the length and regions of “genome coding sections”, as described in Lines 110-112 of the text. What parameters were used to make the tree? Additional details are required here. I assume the scale bar is nucleotide substitutions, but this should be made clear since phylogenetic inferences can also be based on amino acid alignments.
It seems that all boostrap value less than 85 or some cut-off value are “un-supported” and thus the tree would collapse back at those nodes. Can the authors emphasize what regions of the trees are supported – maybe with a color change or bold values?

7. Abstract Line 46-47 The last line reads as follows: “For better
understanding of viral diversity in honey bee colonies, additional metagenomic analyses
are needed.”
It doesn’t seem that “metagenomic analyses” are required to obtain virus sequence information – a targeted virus sequence effort or sequencing of virus enhanced honey bee lysates would likely be better and more cost effective. Since this data is not included in the manuscript, this line should be removed from the abstract.

8. Abstract Line 45 – could be reworded-
“The honey bee virus sequences obtained from bees in Serbia share 93-99% identity with other sequences deposited in GenBank”. Furthermore, this result doesn’t support the idea that additional metagenomic sequencing is required.

9. Line 63 “accused” should be changes to “cited” or “DWV and mites have correlated with honey bee colony losses in multiple studies . .. .”

10. Line 65 – 67. There are better general honey bee viruses citations including
Chen and Siede Honey bee viruses 2007 Ad Virus Research
Brutscher et al. Plos Pathogens
The citations can go at the end of the sentence.

11. Line 68 – Martin citation should also be placed at the end of the sentence – this may be a theme throughout.

12. Line 87-88 Authors should include the sampling dates, which may be important for future studies - since virus prevalence and abundance can change in the same colony on a weekly basis.

13. Results section should include the samples sizes (i.e., prevalence is 74% of XXX number of honey bee samples obtained from XX colonies).

14. Line 47 Is the chi-squared test appropriate here or would a student’s t-test be better?

15. Line 150 – 153 – Why can’t multiple infections be considered? It is quite common that honey bee samples are infected by multiple viruses.

16. Introduction and Discussion could include more citations regarding the number of honey bee colonies lost globally. Perhaps include some recent review articles published in the journal “Current Opinion in Insect Science”.

17. Did all colonies from the same apiary have the same (or more similar) virus profile than those from other apiaries?

18. Could include citations from US (e.g., Cavigli et al Apidologie 2015, and several papers by van Engelsdorp), but the authors may prefer to focus on European countries, which is justifiable.

19. Line 220 – 221 – Not all viruses are transmitted by Varroa destructor (see Glenny et al 2017, LSV2 and mites do not correlate), so it may not be safe to assume that results are due to Varroa management.
This line should be deleted or rephrased.

·

Basic reporting

Tha manuscript in well written, I have two minor remarks before publications:

1. M&M, Lines 90-92: The complete sentence ''The selected colonies....'' should be checked aggain and need to be expalined in another way, because viral infections in bees can not be excluded according to clinical picture.

2. Figure 4: Not necessary-delete it.

Experimental design

no commnet

Validity of the findings

no comment

Additional comments

no comment

Reviewer 3 ·

Basic reporting

The work deals with the prvalence of few honey bee viruses in colonies of different strength situated in teritory of Srbia.

The manuscript is well written and presents good connection between introduction and discussion part taking in account the results obtained.

The manuscript fits weel within the scope of the PeerJ.

Experimental design

The aim of research is in accordance with scope of journal and methods are well described.

Validity of the findings

The data seems well analyzed and the conclusions nice stated and supported by obtained results.

Additional comments

Although, list of references fomatting is not a piority of PeerJ
- references are full and clear,
- list must be unified,
- but style is different than jounal recommand; same in manuscript tekst

Line 74. add reference:
Tlak Gajger, I., J. Bičak, R. Belužić (2014): The occurrence of honeybee viruses in apiaries in the Koprivnica-Križevci district in Croatia. Vet. Arhiv 84, 421-428.

All other corrections and recommendations are marked directly in manuscript (attached to this revision report)

Annotated reviews are not available for download in order to protect the identity of reviewers who chose to remain anonymous.

---

## Round 0.2 · accepted · Accept

Dear Dr. Cirkovic and colleagues:

Thank you for re-submitting your manuscript to PeerJ, and for addressing the concerns raised by the reviewers. I now believe that your manuscript is suitable for publication. Congratulations! I look forward to seeing this work in print, and I anticipate it being an important resource for the honey bee community. Thanks again for choosing PeerJ to publish such important work.

Best,

-joe

#